# Susceptibility to Vaccine-Preventable Diseases in Four Districts of Xaysomboun Province, Lao People’s Democratic Republic

**DOI:** 10.3390/vaccines10030463

**Published:** 2022-03-17

**Authors:** Siriphone Virachith, Mapor Lao, Magnoula Inthepphavong, Saythong Inthalath, Judith M. Hübschen, Sengchanh Kounnavong, Somphou Sayasone, Antony P. Black

**Affiliations:** 1LaoLuxLab, Institut Pasteur du Laos, Vientiane Capital 01030, Laos; s.virachith@pasteur.la; 2Lao Tropical and Public Health Institute, Ministry of Health, Vientiane Capital 01030, Laos; maporlao@gmail.com (M.L.); moly_magnoula@hotmail.com (M.I.); saythong570@gmail.com (S.I.); skounnavong@gmail.com (S.K.); somphou.sayasone@yahoo.com (S.S.); 3Department of Infection and Immunity, Luxembourg Institute of Health, L-4354 Esch-sur-Alzette, Luxembourg; judith.huebschen@lih.lu

**Keywords:** hepatitis B, measles, rubella, tetanus, diphtheria, Laos, vaccines, antibodies

## Abstract

Xaysomboun province has some of the lowest health indicators in Lao People’s Democratic Republic (PDR). This cross-sectional study aimed to determine the vaccination, susceptibility and exposure status of the population to hepatitis B virus (HBV), measles, rubella, and tetanus. Participants aged 5 years and older were randomly selected from four districts. From each enrolled participant, demographic data and 5 mL of blood sample were taken. HBV surface antigen (HBsAg) and antibodies against HBV, measles, rubella, and tetanus were detected by ELISA. A total of 363 participants (age 5 to 80 years) were included. HBV exposure, as determined by anti-HBV core (anti-HBc) antibodies, was 56.2% overall, and was significantly higher among those aged ≥21 years (78.1%). HBsAg was detected in 9.4% overall and increased to 20% in ages 31–40 years. Only 13.8% of participants had serology indicative of vaccination (anti-HBs positive, anti-HBc negative). Seroprotection against measles was 74.6% overall but only 41.7% in children aged 5–10 years. Anti-rubella IgG was 94.2% overall and high in all age groups. Tetanus seroprevalence was only 47.4% overall but significantly higher in females aged 31–40 (75.6%). We suggest strengthening of routine and booster HBV, measles, and tetanus vaccine coverage in Xaysomboun province.

## 1. Introduction

Vaccination is one of the most effective public health interventions worldwide. Since the widespread use of vaccines in the 20th Century, there has been a marked reduction in many previously prevalent infectious diseases such as small pox, polio, measles, rubella, tetanus, diphtheria, and others [1]. Nevertheless, countries with challenged healthcare systems and low vaccination coverage still see outbreaks of vaccine-preventable diseases. Often these same countries have limited disease surveillance capacities. In such instances, serostudies are a powerful and underused tool to assess the susceptibility of populations to infection [2,3].

Lao People’s Democratic Republic (PDR) is a lower middle-income country in the WHO Western Pacific region. Hepatitis B virus (HBV) infection is highly endemic in the country with about 40–50% of the general adult population exposed and 8–10% with chronic infection [4,5,6,7]. Most infection is thought to occur in early childhood, e.g., during birth, breastfeeding, or early life with higher associated risk of developing chronic infection and liver disease [8]. Lao PDR has the 5th highest rate of liver cancer worldwide [9]. Infant vaccination is important to prevent early childhood infection and progression to chronic infection. However, routine screening for hepatitis B surface antigen (HBsAg) in pregnant women and immunoglobulin prophylaxis in newborns are not widely practiced. The HBV vaccine was introduced into the childhood immunization schedule in 2001 at 6, 10 and 14 weeks of age (currently in the form of the pentavalent vaccine; diphtheria-tetanus-pertussis-hepatitis B-hemophilus influezae) and the birth dose was implemented in 2004 [10]. However, there are still large numbers of adults who have not been vaccinated. Furthermore, the coverage of the third dose of the pentavalent vaccine in children aged 12–23 months was only 55.5% in 2011–2012 and 60.8% in 2017. HBV birth dose coverage is also low, with 60.7% in 2017 [11]. Although some impact of the HBV vaccine has been seen in adolescents from Vientiane Capital and Bolhikhamxay province with reduced rates of exposure and chronic infection [12], the levels of infection and vaccination coverage are variable nationwide [13]. It is therefore important to ascertain the levels of HBV infection in particular regions to identify areas of concern and inform the Lao Ministry of Health.

Monovalent measles vaccine was introduced into the routine Lao childhood immunization schedule in 1984 in two provinces and had expanded nationwide by 1994. In 2011, rubella vaccine was included in the immunization schedule and administered as the combined measles/rubella (MR)1 vaccine to infants aged between 9- and 12-month-old. MR2 was included at the age of 12 to 19 months in 2017. However, measles outbreaks were reported in 2012, 2013, and 2014 and coverage of MR1 in 2017 and 2019 was only 66% and 69%, respectively [11,14]. Nationwide measles or MR supplementary immunization activities (SIA) were carried out in 2001, 2007, 2011, 2014, and 2017. In this context, it is important to identify any immunity gaps in the population in order to direct future SIA.

The diphtheria-tetanus-pertussis (DTP) vaccine was gradually introduced into Lao PDR from 1979 and expanded nationwide by 1989 and is now administered as the pentavalent vaccine. DPT3 coverage in 2011–2012 and 2017 was about 55.5% and 60.8%, respectively [11,15,16]. This low vaccine coverage has resulted in several diphtheria outbreaks [17,18] and pertussis is thought to be prevalent [19]. Tetanus infant vaccination is supplemented by vaccination of women of childbearing age (15–49 years old). Together with improved birth and post-natal practices, this vaccination policy resulted in the elimination of maternal and neonatal tetanus (MNT) from Lao PDR in late 2014. Nevertheless, there remain areas for concern, such as missed opportunities to receive vaccination during antenatal care (ANC) [20]. In addition, there are no scheduled routine booster doses for diphtheria and pertussis after infancy and tetanus boosters are limited to women of childbearing age. Therefore, serology data indicating any populations or ages with low immunity are important to guide booster policy.

Xaysomboun Province is a mountainous province located 200 km north of Vientiane Capital with around 85,000 inhabitants. The population is ethnically diverse with Hmong (53.7%); Lao-Tai (19.4%); Mone-Khmer (16.7%) and other ethnicities (10.2%) [21]. It has poor infrastructure and limited health care facilities [22]. In 2017, the neonatal, infant, and under-5 mortality rates in the province were 27, 47 and 51 deaths per 1000 live births respectively and several other health indicators were below the national average. For example, in 2017, despite 66.7% of deliveries taking place at health care facilities, only 29.1% of all newborns received HBV vaccination, compared to 60.7% nationwide. The coverage for some other vaccines in Xaysomboun was the lowest nationwide; DTP3 coverage was 22.4%, compared to 60.8% nationwide and MR1 coverage was 39.7%, compared to 66.0% nationwide. Estimates of protection against tetanus in women 15–49 years of age in 2017 were also the lowest nationwide at 17.8%, compared to 48.9% [11].

The aim of the current study was to characterize age-specific vaccination and/or exposure to HBV, measles, rubella and tetanus of the population in Xaysomboun and to provide evidence-based data to inform vaccination policy and other control measures.

## 2. Materials and Methods

### 2.1. Study Population

This cross-sectional study was conducted between February and March 2020 in Xaysomboun province. Five villages in each district (Anouvong, Longcheng, Thathom, Longxan and Hom) were randomly selected using Excel “RAND” function. In each village, a list of villagers (anonymized with unique identifiers) was obtained and 20–25 persons aged 5 years and older were randomly selected as above. Prior to enrolment, a signed informed consent form was taken from the participant after reading the information sheet. Healthcare staff read to illiterate participants. A fingerprint and signature of a witness were taken for those unable to sign. For participants under the age of 18, parents or caretakers gave informed consent. In addition, those aged 15–17 gave written assent to take part in the study. 

The village office or health center was selected as the station for the research team. Participants were invited to these stations for an interview to collect socio-demographic data (age, sex, place of birth, ethnicity, religion, occupation, level of education, family income; see Table 1). For participants aged less than 15 years old, the child demographics were given by their parents or caretakers. Furthermore, a blood sample of 5 mL was drawn from each participant and centrifuged to collect the serum. A rapid diagnostic test for HBsAg was done on the spot and the result was given back to the participants. The rest of serum was transferred to Institut Pasteur du Laos for serological investigation on HBV, measles, rubella, and tetanus using ELISA techniques. 

The sample size was calculated using a precision of 5% and an estimated prevalence of anti-tetanus antibody of 50% which was based on previous studies of women [20], healthcare workers [6] and children [23] in four provinces. Using the formula *N* = $\frac{{\left(1.96\right)}^{2}\times 0.5\times \left(1-0.5\right)}{{\left(0.05\right)}^{2}}$ the minimum sample size was 423.

### 2.2. Serology Testing

Anti-hepatitis B core and surface antibodies (anti-HBc and anti-HBs) were detected by commercial ELISA (Diasorin, Italy) and all anti-HBc positive/anti-HBs negative samples were tested for HBsAg (Diasorin, Italy), which would account for the majority of HBsAg positive participants [4]. HBsAg positive samples were defined as “acute or chronic infection”, anti-HBc positive as “previously exposed” and anti-HBc negative, anti-HBs positive as “previously vaccinated”. 

Measles, rubella, and tetanus ELISA results were interpreted according to the manufacturer (Euroimmun, Germany). Thus, anti-measles IgG titers <200 IU/L were considered negative, titers between 200 and <275 IU/L as borderline, and titers ≥275 IU/L as positive. Anti-rubella IgG titers <8 IU/mL were considered negative, titers between 8 and <11 IU/mL as borderline, and a titers ≥11 IU/mL as positive. 

For anti-tetanus IgG, titers <0.1 IU/mL were defined as “insufficient immunity: immediate boost recommended”, those from 0.1 to 0.5 IU/mL as “low immunity: immediate boost recommended”, >0.5 to 1.1 IU/mL as “sufficient immunity: booster recommended in 2–5 years”, >1.1 to 5.0 IU/mL as “sufficient immunity: booster recommended in 5–10 years” and those above 5.0 IU/mL: “sufficient immunity: booster recommended in approximately 10 years”. Protective anti-tetanus antibody titers were defined as those above 0.5 IU/mL.

### 2.3. Statistical Analysis

The completeness and correctness of data were checked before data analysis. All data were analyzed with STATA version 14. Descriptive data were calculated for categorical variables and the means with minimum, maximum, and standard deviation (SD) for continuous variables.

Bivariate analysis was applied to determine the association between the independent variables (HBV, tetanus, measles, and rubella serostatus) and the dependent variables (sociodemographics). All variables with *p* value ≤ 0.2 in bivariate model were included for multivariable analysis. Adjusted odds ratios (aOR) were described. 

The study was conducted according to the guidelines of the Declaration of Helsinki, and approved by the Internal Review Board of Institut Pasteur du Laos and the Lao National Ethics Committee for Health Research (Ref. number 010/NEHCR/2020).

## 3. Results

### 3.1. Population Characteristics

Twelve people were excluded due to refusal to participate from fear of needles and were replaced by 12 other randomly selected participants. Six participants withdrew from the study after consent, due to fear of the blood draw, and one withdrew due to difficulty to draw blood. Due to COVID-19 restrictions, the study could not be conducted in Anouvong District. Thus, the total number of participants was 363. The median age was 25 years (range 5 to 80), 61.2% were female and 42.9% were Hmong. Most participants followed animism (66.9%) and 57.3% were married. 78.5% of the participants were born at home and most (82.9%) had a family monthly income of 1 million to 3 million kips (Table 1).

### 3.2. Hepatitis B Virus (HBV)

Anti-HBc antibody seroprevalence, indicating HBV exposure, was 204/363 (56.2%) overall. There was no significant difference between male and female participants. Exposure was 10/72 (13.9%) in the 5–10-year-old age group but was found to be significantly higher in all older age groups after multivariable analysis, with the highest at 39/45 (86.7%) in those aged 31–40 years (aOR 21.33 [5.26–86.53] *p* < 0.001). Participants from Thathom district had higher anti-HBc seroprevalence than those from Longxarn (61.9% and 47.3%; aOR 3.16 [1.54–6.49] *p* = 0.002) (Table 2 and Appendix A). 

Anti-HBc positive/anti-HBs negative samples were screened for HBsAg. Assuming all other samples were HBsAg negative [4], the overall prevalence of HBsAg, indicating chronic (or acute) infection was 34/363 (9.4%). Males were found to have higher seroprevalence of HBsAg (17/141, 12.1%) than females (17/222, 7.6%; aOR 2.22 [1.03–4.76], *p* = 0.04). HBsAg seroprevalence was 1/72 (1.4%) in children aged 5–10 years and it increased to 7/66 (10.6%, aOR 11.98 [1.38–103.87], *p* = 0.02) in age group 21–30 years, 9/45 (20.0%, aOR 30.39 [3.48–264.98], *p* = 0.002) in 31 to 40 year olds, and 10/95 (10.5%, aOR 10.11 [1.23–82.46] *p* = 0.03) in those aged 40 years or older. In addition, participants from Thathom district had a higher HBsAg seroprevalence (16/105, 15.2%) compared with Longxarn (6/93, 6.4%; aOR 3.15 [1.13–8.79] *p* = 0.02]) (Figure 1, Table 2 and Appendix A).

The overall prevalence of serology indicating vaccination (anti-HBs positive, anti-HBc negative) was 50/363 (13.8%) (Table 2 and Appendix A). After multivariable analysis, males had a higher prevalence of serological vaccination profile (29/141, 20.6%) than females (21/222, 9.5%; aOR 2.06 [1.10–3.86], *p* = 0.02). Vaccination serology in those aged 5–10 years old was 21/72 (29.2%) but was significantly lower in those aged 21–30 years (6/66, 9.1%; aOR 0.25 [0.09–0.68], *p* = 0.007) and those aged over 40 years (8/95, 8.4%, aOR 0.23 [0.09–0.57], *p* = 0.001) (Appendix A and Figure 1). 

### 3.3. Measles and Rubella

Overall, 220/363 (60.6%) participants were seropositive for anti-measles IgG antibodies and 51/363 (14.1%) were borderline, with no significant difference between males and females. Measles seroprevalence in those aged 5–10 years was 18/72 (25.0%) but was significantly higher in adults aged 21–30 years old (48/66, 72.7%; aOR 3.47 [1.20–10.06], *p* = 0.02), those aged 31–41 years (41/45, 91.1%; aOR 10.44 [2.45–44.42], *p* = 0.001) and those aged above 40 (86/95, 90.5%, aOR 10.48 [2.97–37.02], *p* < 0.001) after multivariable analysis. Similarly, the highest antibody titers against measles were found in those 31–40 years of age (median = 1301 IU/mL) compared to those 5–10 years (median = 183.2 IU/mL, *p* < 0.0001) and 11–20 years (median = 220.7 IU/mL, *p* < 0.0001) (Figure 2). Married participants had higher anti-measles antibody seroprevalence (176/208, 84.6%) than unmarried participants (44/155, 22.4%; aOR 2.7 [1.09–37.02], *p* = 0.03) (Appendix A). 

The majority of participants had high seroprevalence of anti-rubella antibodies, with 331/363 (91.2%) and 11/363 (3.0%) seropositive and borderline, respectively. Although no factors were significantly associated with anti-rubella seropositivity following multivariable analysis, the 11–20-year-old participants had significantly higher median anti-rubella antibody titers (63.7 IU/mL) than 5–10 year-olds (median = 39.7 IU/mL, *p* < 0.001) (Figure 2 and Appendix A).

### 3.4. Tetanus

Overall tetanus serology indicated that 136/363 (37.5%) had “insufficient immunity”, and 55/363 (15.1%) had “low immunity”. “Sufficient immunity” (>0.5 IU/mL) was detected in 172/363 (47.4%), with 79/363 (21.8%), 85/363 (23.4%), and 8/363 (2.2%) needing future booster doses in 2–5, 5–10, and 10 years, respectively. Males had significantly lower prevalence of sufficient immunity (29/141, 20.6%; aOR 0.12 [0.07–0.21], *p* < 0.001) than females (143/222, 64.1%). Susceptibility to tetanus in male adults aged 21–30 and 31–40 years old (20/25, 80% and 8/9, 88.9%) was higher than in females from the same age groups (3/41, 7.3%; *p* < 0.0001 and 1/36, 2.8%; *p* < 0.0001) (Figure 3 and Appendix A).

Sufficient immunity to tetanus increased significantly in those aged 11–20 years (39/85, 45.9%; aOR 2.10 [0.99–4.43], *p* = 0.05); those aged 21–30 years (35/66, 53.0%; aOR 2.65 [1.17–6.00], *p* = 0.01) or those aged 31–40 years (34/45, 75.6%; aOR 6.47 [2.48–16.85], *p* < 0.001), compared to those aged 5–10 years (23/72, 31.9%). Buddhists had higher tetanus seroprotection (74/120, 61.7%; aOR 2.8 [1.40–5.67], *p* = 0.003) than Animists (98/243, 40.3%). Seroprotection was significantly higher in participants from Hom district (40/83, 48.2%; aOR 2.37 [1.16–4.86] *p* = 0.01) and Thathom district (62/105, 59.1%; aOR 2.6 [1.19–5.65] *p* = 0.01) compared to those living in Longxarn district (34/93, 36.6%) (Appendix A).

## 4. Discussion

Xaysomboun province is one of the smallest provinces in Lao PDR, with some of the poorest health indicators in the country, including the lowest reported vaccination coverage for measles and lowest tetanus protection in women. This study aimed to determine the serological profile of HBV, measles, rubella, and tetanus at different ages in Xaysomboun province.

HBV serology indicative of vaccination (anti-HBs positive/anti-HBc negative) was only 29.2% among those aged 5–10 years, decreasing to 17.6% in 10–20 years of age, similar to our previous findings in other provinces [12,24]. These low levels likely reflect low vaccination coverage in addition to waning of antibodies and/or low vaccine immunogenicity. Importantly, vaccinated individuals without anti-HBs antibodies may still be protected by B cell memory and other mechanisms [25]. Indeed, there was a significant decrease in HBV exposure and chronic infection in those aged less than 11 years, reflecting a positive impact of introducing routine HBV immunization in Lao infants since 2001. Furthermore, we found only 1.4% HBsAg seroprevalence in children aged 5–10 years, similar to some previous reports in Lao PDR [24] but lower than others [5]. A similar decrease in HBV exposure and chronic infection has been seen in previously endemic countries such as Taiwan and Thailand, following the introduction of the HBV vaccine for newborns and infants [26]. Compared to our previous study in Saravan, adolescents aged 11–20 years had a similar but slightly higher rate of HBV exposure (4.3% versus 2.8%) [27], possibly reflecting a later rollout of vaccination in Xaysomboun province at the beginning of HBV vaccine introduction. Our study also found that women between the ages of 21 and 40, when child-birth is common, had a high HBsAg infection rate (12.2–16.7%), emphasizing the importance of the HBV vaccine 24 h after birth and facility-based deliveries. Facility-based births and timely HBV birth-dose vaccination have been shown to be particularly challenging in rural areas of Lao PDR [11,28]. The higher infection rate in males than in females, especially in the older age groups, is seen in other studies in Lao PDR [4,5,6,7] and elsewhere [29,30], although the reasons are unknown and could include biological as well as societal factors. The reason why Thathom district had a higher percentage of HBsAg (15.2%) than other districts is unknown but could include specific risk factors e.g., tattooing. This requires further investigation.

Seroprotection against measles was as high as 74.6%, but nevertheless not high enough to reach herd immunity (93–95%) [31]. In children aged 5–10 years, measles seroprotection was only 41.7%, which is in line with only 37.9% MR1 coverage rate of children aged 12–23 months in Xaysomboun province in 2017 [11] and similar to our previous study in the south of Lao PDR [27]. Similarly low levels in other settings are associated with high risk of measles outbreaks [3]. Higher titers and seroprevalence in older age groups could be due to wild type exposure or SIA which used monovalent or bivalent (MR) vaccine since 2001 until 2017. Although these SIA only targeted people as young as 9 months up until the ages of 19 years, the older participants in our study would have had more chance to be vaccinated over the years [14]. These data, together with the low routine infant vaccination coverage, indicate a need for further SIA in young age groups in addition to strengthening of the routine MR1 and MR2 vaccination. 

Seroprotection against rubella in our study was higher than measles in all age groups with 91.2% overall. This is similar to other provinces [27] and high seroprevalence in children could reflect wild-type rubella circulation [32], or vaccination—suggesting a difference in immunogenicity between measles and rubella components of the MR vaccine [14,33]. In contrast, the high seroprevalence in adults born before introduction of rubella vaccination in 2011 is likely to be due to rubella virus infection or SIA for younger adults.

Overall seroprotection against tetanus was 47.4%, and was significantly higher in females (64.1%) than in males (20.6%, *p* < 0.001). The high protection of females, particularly those aged 31 to 40 (97.2%), likely results from the policy to vaccinate women of child bearing age (15–49 years old). The high susceptibility of males to infection, similar to other countries [34,35], is concerning. Although there are no published data on tetanus cases in Xaysomboun province, assessment of the tetanus burden among the different sexes is warranted, with a view to strengthening vaccination and other control measures e.g., raising awareness, in males. The lower seroprotection in those who follow animism, as compared to Buddhism (*p* = 0.003) is difficult to explain and warrants further investigation. It may reflect a distrust of animists for non-traditional medicine and/or low ANC attendance during pregnancy. Lastly, different levels of susceptibility between the districts may reflect variable vaccine coverage. The levels of tetanus antibodies that we find are low in comparison to neighboring countries, particularly in adolescents and adults. In Thailand, 99% of adolescents aged 11–20 years had protective tetanus antibodies, compared to 45.9% in the current study [36]. The higher antibody levels in Thailand likely results from the policy to have additional DTP boosters for males and females at 18 months and 4–6 years of age, a policy that is not implemented in Lao PDR.

A limitation of our study is that we did not assess the determinants of vaccination. It is possible that low seroprevalence was due to low vaccination coverage (logistical constraints of the vaccine chain, or low compliance with vaccination [37,38]) or poor quality vaccines [23]. Such reasons are important to assess in more detail in order to reduce the vulnerability to infection in the population. In addition, due to COVID-19 restrictions it was not possible to recruit participants from one of the target districts. As a consequence, the minimum sample size was not reached and therefore the power of the statistical analysis may have been reduced.

## 5. Conclusions

The high levels of HBV exposure and of chronic infection indicate that there will be a burden of liver disease in the future. Focusing on HBV vaccination at birth, facility-based deliveries are warranted. Although rubella seropositivity was high at all ages, measles seropositivity and antibody titers were low in young children. Further SIA should be considered, focusing on the young age groups, in addition to strengthening of routine infant vaccination. Lastly, although the overall seroprotection of females against tetanus was high, there is room for improvement in adolescents. Low seroprotection of males against tetanus may warrant booster doses. 

## Figures and Tables

**Figure 1 vaccines-10-00463-f001:**
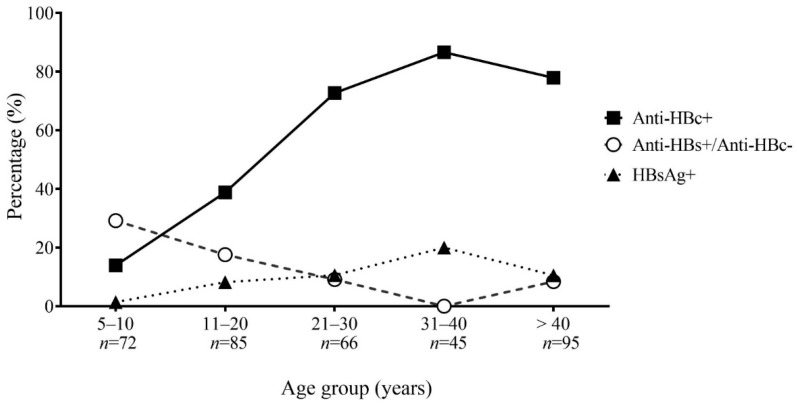
Age-stratified Hepatitis B serology.

**Figure 2 vaccines-10-00463-f002:**
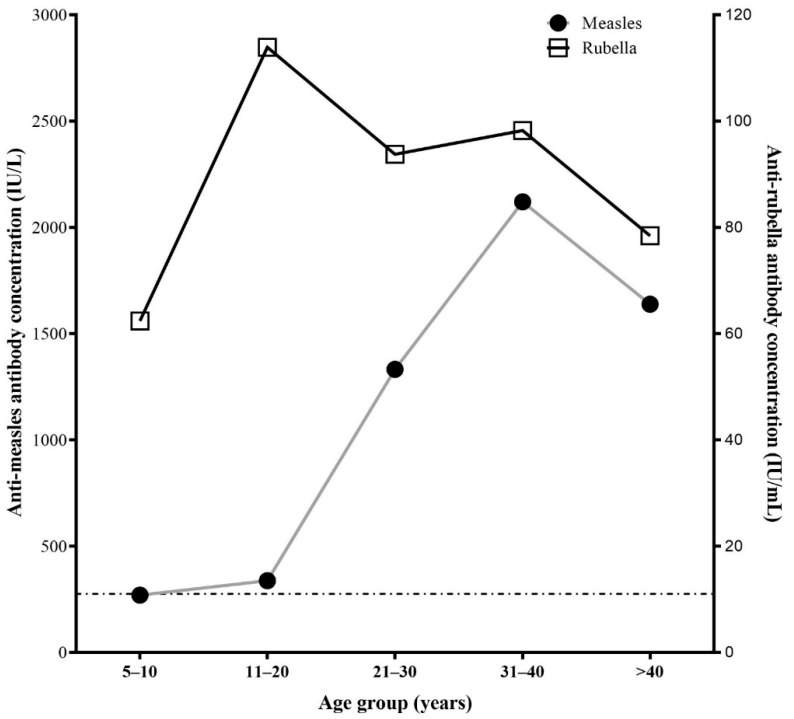
Age-stratified anti-measles and anti-rubella median antibody titers by age group. The horizontal dotted line represents the positivity cut-off for anti-measles (275 IU/L) and anti-rubella (11 IU/mL) antibodies.

**Figure 3 vaccines-10-00463-f003:**
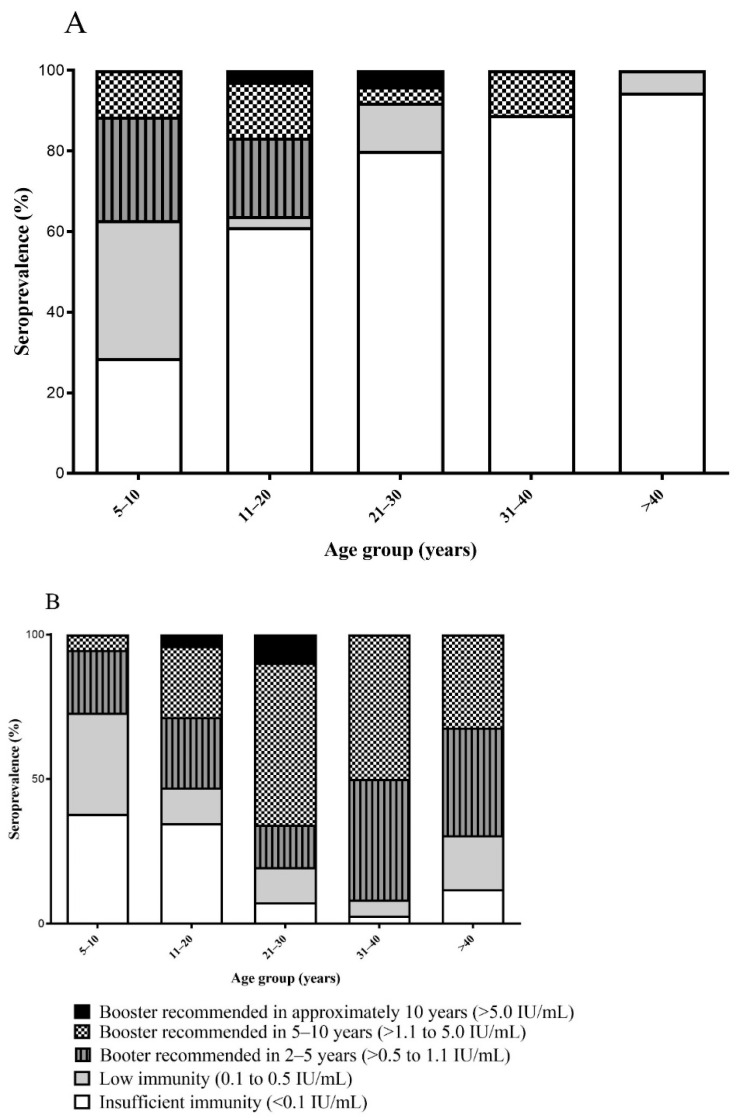
Age-stratified tetanus seroprevalence status according to sex. (**A**) Male; (**B**) female.

**Table 1 vaccines-10-00463-t001:** Socio-demographic characteristics of study participants, Lao PDR (*n* = 363).

Characteristics		N (%)
Sex	Female	222 (61.2)
	Male	141 (38.8)
Age groups (years)	5–10	72 (19.8)
	11–20	85 (23.4)
	21–30	66 (18.2)
	31–40	45 (12.4)
	>40	95 (26.2)
Ethnicity	Lao-Tai	141 (38.8)
	Hmong-Mien	156 (42.9)
	Mone-Khmer	66 (18.2)
Religion	Animist	243 (66.9)
	Buddhist	120 (33.1)
District	Longxarn	93 (25.6)
	Hom	83 (22.9)
	Longchaeng	82 (22.6)
	Thathom	105 (28.9)
Place of birth	Provincial hospital	18 (4.9)
	District hospital	41 (11.3)
	Health center	19 (5.2)
	At home	285 (78.5)
Marital status	Single	155 (42.7)
	Married	208 (57.3)
Education	None	118 (32.5)
	Elementary	123 (33.9)
	Secondary/high school	107 (29.5)
	University	15 (4.1)
Family monthly income (Kip) *	<1,000,000	33 (9.1)
	1,000,000–3,000,000	301 (82.9)
	>3,000,000	29 (8.0)

* 1000 Kip = approximately 0.104 USD.

**Table 2 vaccines-10-00463-t002:** Hepatitis B virus serology by age group and sex. Anti-HBc positive/anti-HBs negative samples were screened for HBsAg. All other samples were assumed to be HBsAg negative [4].

			Age Group (years)										
Anti-HBs	Anti-HBc	HBsAg	5–10		11–20		21–30		31–40		>40		Total
			Male	Female	Male	Female	Male	Female	Male	Female	Male	Female	
-	-	-	21/35 (60.0)	20/37 (54.1)	11/36 (30.6)	26/49 (53.1)	3/25 (12.0)	9/41 (21.9)	0/9 (0)	6/36 (16.7)	3/36 (8.4)	10/59 (16.9)	109/363 (30.1)
+	-	-	10/35 (28.6)	11/37 (29.7)	9/36 (25.0)	6/49 (12.2)	4/25 (16.0)	2/41 (4.9)	0/9 (0)	0/36 (0)	6/36 (16.6)	2/59 (3.4)	50/363 (13.8)
+	+	-	3/35 (8.6)	5/37 (13.5)	8/36 (22.2)	11/49 (22.4)	16/25 (64.0)	18/41(43.9)	6/9 (66.6)	18/36 (50.0)	12/36 (33.4)	24/59 (40.7)	121/363 (33.3)
-	+	-	1/35 (2.8)	0/37 (0)	2/36 (5.5)	5/49 (10.2)	0/25 (0)	7/41 (17.1)	0/9 (0)	6/36 (16.7)	9/36 (25.0)	19/59 (32.2)	49/363 (13.5)
-	+	+	0/35 (0)	1/37 (2.7)	6/36 (16.7)	1/49 (2.1)	2/25 (8.0)	5/41 (12.2)	3/9 (33.4)	6/36 (16.7)	6/36 (16.6)	4/59 (6.8)	34/363 (9.4)

## Data Availability

The anonymized data are available upon reasonable request from the corresponding author.

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
