# Peer review of "Susceptibility to Vaccine-Preventable Diseases in Four Districts of Xaysomboun Province, Lao People’s Democratic Republic"

_vaccines, 2022, doi:10.3390/vaccines10030463_

Round 1
Reviewer 1 Report
Thank you for asking me to review this article. The ongoing pandemic has resulted in global health, economic and social crises. The risk of forgiving other vaccine preventable diseases is real. In particular, low and middle income countries may suffer from lack of preventive strategies for high burden infectious diseases. In this context, the paper under evaluation is aimed at determinating the vaccination, susceptibility and exposure status of the population to hepatitis B virus, measles, rubella and tetanus in Lao Democratic Republic.
The subject under study is certainly very important, especially in the historical period we are experiencing. The article presents interesting results but, I would like to encourage authors to consider several issues to be improved. I believe that after incorporating these issues, the paper will have a value for this journal. I hope that my comments are useful for authors, as they further develop the manuscript.
Title: it is overstated since it is carried on in a sample of people.
Introduction: The authors should better clear what is the gap in the literature that is filled with this study. The paper is too much targeted at a local level. Finally, the Authors must explicit what is the potential contribution of the study to the literature and its implications.
Methods: The enrolment procedure must be better specified. How did the authors choose the way used to enroll their random sample? How did they avoid the selection bias? The authors calculate the minimum sample size, but what is the reference population? All country population? The acquisition a valid informed consent must be better detailed, especially under 18 years. The selected variables to be investigated must be described and also how the authors selected them. A questionnaire was used? Was it validated?
Statistical analysis: I suggest to insert a measure of the magnitude of the effect for the comparisons, include an effect sizes analysis.
Discussion: It is sometimes redundant, it should be reorganized emphasizing the contribution of the study to the literature, discussing results in an international context. Moreover it is demonstrated that understanding the determinants of vaccination acceptance and coverage is a useful tool for implementing strategic measures aimed at improving patient compliance with vaccination; this study do not consider this important issue. Therefore, the authors must clearly discuss the impact of vaccine acceptance and information campaign on the level of vaccine coverage in the limit section (refer to article with doi: 10.3390/vaccines9060638).
Reviewer 2 Report
This is a cross-sectional study aiming to determine the vaccination, susceptibility and exposure status of the population of Xaysomboun Province of Lao PDR. This is a seroprevalence study age stratified that provides recommendations in the public health field. The paper is of interest for the readers and for the journal; however, several changes should be made before publishing it.
The introduction section is mainly focused on the characterization of the population of Lao People's Democratic Republic (PDR). I would also introduce, at the beggining of the paper, the main advantages and impact of vaccination in earlier ages in public health, worldwide and also in lower middle-income countries.
The main aim of this study is not adequately described. At the end of the introduction section the authors report that they "carried out an age-stratified seroprevalence study" but they did not describe the main goals of the present work.
Sample size calculation is described at the beginning of the methods section. I would prefer to better describe the study design, population, and then the sample size.
Statistical analyses is preferred to "data analysis".
Sections and subsections of the manuscript should be renumbered.
Table 5 is very extensive. I would prefer to present it as a Supplementary material.
The conclusions section is long and does not represent a summary. I would rename it as Discussion section and then build a conclusions section.
Reviewer 3 Report
Estimated Authors,
I've been asked to perform a review of the revised version of the paper "Susceptibility to vaccine-preventable diseases in four districts of Xaysomboun Province, Lao People’s Democratic Republic".
This seroprevalence study, performed in a specific Province of Lao PDR (as explained by the Authors, Xaysomboun province is both geographically remote as a montainous area, and economically deprived), identified interesting vaccination rates: 13.8% for HBV, 74.6% for measles but only 41.6% in children, tetanus 47.4%. In other words, such results urge for substantial interventions aimed to improve vaccination rates for the aforementioned communicable disorders, and also stress how difficult may be achieve high vaccination rates for new vaccinations (e.g. SARS-CoV-2) if required by new and changing conditions.
In my opinion, the present paper may be accepted, and the Authors have radically improved the text and data reporting from the original text that was originally reviewed by different professionals. However some minor improvements are still required:
1) the discussion should cope with the relatively reduced number of sampled individuals (according to the power analysis more than 400 sampled had been required, but authors were able to recruit only around 360);
2) please double check the text for some minor remaining typos.
Author Response
Thank you for reviewing our revised manuscript and your further suggestions, which we have addressed as follows
- the discussion should cope with the relatively reduced number of sampled individuals (according to the power analysis more than 400 sampled had been required, but authors were able to recruit only around 360);
We have now added a section to the “limitations paragraph” as follows:
“In addition, due to COVID-19 restrictions it was not possible to recruit participants from one of the target districts. As a consequence, the minimum sample size was not reached and therefore the power of the statistical analysis may have been reduced.”
- 2) please double check the text for some minor remaining typos.
We have corrected several typos in the text and left as “tracked changes”
Round 2
Reviewer 1 Report
Although the authors have done a good job in improving the manuscript, in my opinion there are problems in the research design, which cannot be improved and which seem to me to be particularly serious.
I hope that these comments will not discourage the authors and that they will take them as an impulse to improve.
Author Response
Thank you for your comment.
We are sorry that you did not find our responses to your first suggestions appropriate as we believed that we addressed all of the points.
We also note that your assessment of the manuscript has decreased since the first version although we believe that the manuscript has improved, thanks to your many important previous suggestions.
Once again, thank you for taking the time to read and comment on our manuscript.
Reviewer 2 Report
The authors have improved the manuscript. I consider that it can be published in its current form.
Author Response
Thank you for your review and reading our revised manuscript